# Gaseous Elemental Mercury Exchange Fluxes over Air-Soil Interfaces in the Degraded Grasslands of Northeastern China

**DOI:** 10.3390/biology10090917

**Published:** 2021-09-15

**Authors:** Gang Zhang, Xuhang Zhou, Xu Li, Lei Wang, Xiangyun Li, Zheng Luo, Yangjie Zhang, Zhiyun Yang, Rongfang Hu, Zhanhui Tang, Deli Wang, Zhaojun Wang

**Affiliations:** 1School of Environment, Northeast Normal University, Changchun 130117, China; zhangg217@nenu.edu.cn (G.Z.); zhouxh561@nenu.edu.cn (X.Z.); lix896@nenu.edu.cn (X.L.); wangl788@nenu.edu.cn (L.W.); lixy869@nenu.edu.cn (X.L.); luoz932@nenu.edu.cn (Z.L.); zhangyj415@nenu.edu.cn (Y.Z.); yangzy295@nenu.edu.cn (Z.Y.); hurf556@nenu.edu.cn (R.H.); tangzh789@nenu.edu.cn (Z.T.); z35976113@163.com (D.W.); 2Key Laboratory of Vegetation Ecology, Ministry of Education, Northeast Normal University, Changchun 130117, China; 3Institute of Grassland Science, Northeast Normal University, Changchun 130117, China; 4State Environmental Protection Key Laboratory of Wetland Ecology and Vegetation Restoration, Changchun 130117, China

**Keywords:** dynamic flux chamber, mercury fluxes, soil/air interfaces, saline and alkaline land types, vegetation

## Abstract

**Simple Summary:**

This study investigated the gaseous elemental mercury exchange fluxes over *Artemisia anethifolia* coverage and removal and bare soil using a dynamic flux chamber attached to the Lumex^R^ RA915+ Hg analyzer during the growing season from May to September of 2018, in which the interactive effects of plant coverage and meteorological conditions were highlighted. The results showed that the net emissions from the soil to the atmosphere, which varied diurnally, with releases occurring during the daytime hours and depositions occurring during the nighttime hours. Significant differences were observed in the fluxes between the vegetation coverage and removal during the growing months. In addition, it was determined that the mercury fluxes were positively correlated with the solar radiation and air/soil temperature levels and negatively correlated with the air humidity and soil moisture. The grassland soil served as both a source and a sink for atmospheric mercury, depending on the season and meteorological factors. The plants played an important inhibiting role in the mercury exchanges between the soil and the atmosphere. This research will potentially assist in the development of more accurate local and regional estimates of mercury emissions from degraded grassland areas and the terrestrial environment as a whole.

**Abstract:**

Mercury (Hg) is a global pollutant that may potentially have serious impacts on human health and ecologies. The gaseous elemental mercury (GEM) exchanges between terrestrial surfaces and the atmosphere play important roles in the global Hg cycle. This study investigated GEM exchange fluxes over two land cover types (including *Artemisia anethifolia* coverage and removal and bare soil) using a dynamic flux chamber attached to the Lumex^R^ RA915+ Hg analyzer during the growing season from May to September of 2018, in which the interactive effects of plant coverage and meteorological conditions were highlighted. The daily mean ambient levels of GEM and the total mercury concentrations of the soil (TSM) were determined to be 12.4 ± 3.6 to 16.4 ± 5.6 ng·m^−3^ and 32.8 to 36.2 ng·g^−1^, respectively, for all the measurements from May to September. The GEM exchange fluxes (ng·m^−2^·h^−1^) during the five-month period for the three treatments included the net emissions from the soil to the atmosphere (mean 5.4 to 7.1; range of −27.0 to 47.3), which varied diurnally, with releases occurring during the daytime hours and depositions occurring during the nighttime hours. Significant differences were observed in the fluxes between the vegetation coverage and removal during the growing months (*p* < 0.05). In addition, it was determined that the Hg fluxes were positively correlated with the solar radiation and air/soil temperature levels and negatively correlated with the air relative humidity and soil moisture under all the conditions (*p* < 0.05). Overall, the results obtained in this study demonstrated that the grassland soil served as both a source and a sink for atmospheric Hg, depending on the season and meteorological factors. Furthermore, the plants played an important inhibiting role in the Hg exchanges between the soil and the atmosphere.

## 1. Introduction

Mercury (Hg) is considered to be a significant global pollutant due to its biochemical properties [1]. The main concerns at present regarding Hg are related to its mobility and persistence [2], as well as its bioaccumulation through the trophic web [3]. Gaseous mercury, which comprises over 95% of the total Hg in the air [4], has the ability to exist in the atmosphere for more than one year. In addition, it can travel long distances along with the atmospheric circulation, resulting in global Hg contamination [1,5]. Therefore, research investigations regarding the sources of Hg are essential in order to deepen the understanding of the biogeochemical cycle of Hg on a global scale [6]. Early studies have estimated that as much as 80% of the Hg deposited on terrestrial surfaces will be re-emitted to the atmosphere through surface emissions [7]. Those results have further indicated the importance of terrestrial environments as significant Hg sources in the global Hg cycle. The gaseous mercury exchange fluxes between atmospheric and terrestrial sources serve as important examples. However, those fluxes have been poorly characterized, and the routes by which Hg enters and exits the Earth’s different ecosystems require further clarification [8]. It is expected that by improving the current quantification of fundamental ecosystems, accurate predictions can be made regarding the impacts that climate change will have on the global Hg cycle [9].

Many of the previous research attempts to quantify or mechanistically understand gaseous mercury fluxes were focused on forests [10,11], wetlands [12], and croplands [13]. Unfortunately, at the present time, relatively little attention has been given to grasslands, which are the largest terrestrial ecosystem type. Grasslands directly contribute to such livestock production industries as dairy, wool, and leather, which support almost a billion people worldwide [14]. Howard et al. [15] made a short-term measurement of the Hg fluxes from Australian alpine grassland soil. However, available data from long-term measurements and different types of grassland areas remain scarce. 

The Hg fluxes between soil surfaces and the atmosphere have been extensively studied in terrestrial ecosystems during the past thirty years and have been estimated to contribute 256 to 1400 Mg·y^−1^ worldwide [16,17]. Different terrestrial ecosystems have their specific functions as sink pools or net sources of mercury. For example, forests are generally regarded as active pools of Hg. Canopy foliage uptake the majority of the Hg from atmospheric sources through stoma rather than root uptake [18,19]. These fixed Hg can be stored in live biomass (for example, stems) and then transported to the forest floor through litterfall and throughfall. The mercury will then be sequestrated in the soil [20]. The vegetation canopies, which absorb up to 99% of solar radiation, are expected to reduce Hg^0^ emissions by limiting the warming of the underlying soil. An earlier estimate of net Hg^0^ evasion from forest soils was approximately 340 Mg·y^−1^ [21]. Unlike the forests, wetlands are both particularly important sinks of atmospherically deposited Hg and emission sources [6]. Atmospheric Hg deposited into wetlands via wet and dry mechanisms may easily be trapped. This is due to the fact that wet sediment tends to host more dissolved organic carbon, which can solubilize Hg [22]. The majority of wetlands in the background sites are relatively small pools of the environmental Hg. Meanwhile, in Hg-enriched regions, they tend to be more significant emission sources to the atmosphere when compared with forest soils due to adequate solar radiation levels, which enhance the Hg^2+^ reduction to Hg^0^. The few studies which have been completed which involved grassland regions only focused on the Hg exchange fluxes between frozen layer soil and the atmosphere in alpine or high-elevation regions [15,23,24,25]. The results of those investigations revealed that the alpine grassland soil serves as a weak source of Hg released into the atmosphere but has become gradually stronger with recent climate warming trends. Meanwhile, the Hg^0^ fluxes presented obvious diurnal and seasonal variabilities, with emissions generally occurring during daytime hours and warmer seasons. In addition, depositions were observed to occur during nighttime hours and colder seasons. However, no clear agreement has been reached on the aforementioned characteristics [26]. In summary, the large uncertainties caused by different measurement methods, along with a lack of knowledge of soil-air Hg exchanges in grassland regions (particularly whole ecosystem soil-plus-vegetation flux data), have resulted in substantial controversy regarding the roles played by global grassland surfaces. Therefore, a definite gap in understanding currently exists in regard to the specific roles of grasslands in regional and global Hg biochemical cycles [6].

China’s temperate steppe areas are considered to be the third-largest distribution area of grasslands in the world and one of the most widely distributed vegetation types on Earth [27]. Under the influencing effects of human activities, the land usage and coverage of the temperate steppe areas in China have undergone significant changes in recent decades. For example, large areas of grassland have been reclaimed for farmland usages, and overgrazing and expansion of residential land have resulted in serious degradation of grassland regions [28]. Grassland degradation is mainly manifested by the thinning of low grassland vegetation and the reduction in surface exposure due to litter coverage. The results of this research will potentially assist in the development of more accurate local and regional estimates of Hg emissions from degraded grassland areas and the terrestrial environment as a whole.

## 2. Materials and Methods

### 2.1. Study Area

The study area was located in Changling County, southwestern section of the Songnen Plains, within the eastern region of the Eurasian steppes (44°40′–44°44′ N, 123°44′–123°47′ E) at an altitude range of 110 to 180 m (Figure 1). The site was characterized by a semi-arid continental monsoon climate, with cold, dry winters and warm, rainy summers. The annual mean temperature ranged from 4.6 to 6.4 °C. The annual precipitation ranged between 280 and 400 mm, with 70% of the precipitation falling between the months of June and August [29]. The main soil type was characterized by high salinity and alkalization (pH 8.3 to 10.0). *Artemisia anethifolia*, Kochia scoparia, and Artemisia scoparia were determined to be the common species at the alkaline sites. *Artemisia anethifolia* is the most salinity-tolerant Artemisia plant, which forms small communities on meadow grassland and dry grassland. The increase in this artemisia is often a sign of overgrazing or grassland degradation. In addition, the growing season is from May to September each year, which is the golden period for field experiments in Northeast China. So, we selected *Artemisia anethifolia* as our experimental species. Two types of land cover were selected within the study area: *A**. anethifolia* and bare soil. The treatments included the original and mowing of the above-ground plants.

### 2.2. Sampling Site and Experimental Design

This study’s experiments were carried out in a 25 × 25 m^2^ flat and enclosed meadow steppe. Good vegetation uniformity was observed, with *A**. anethifolia*, K. scoparia, and A. scoparia determined to be the common species at the alkaline sites. Two 0.25 × 0.40 m^2^ random plots were defined during each monthly period on the basis of species composition as *A**. anethifolia* and bare soil.

The experimental design included randomized blocks which were established monthly during the growing season (May to September) in 2018. This study’s sampling sites were located within two blocks (*A**. anethifolia* and bare soil), with two levels of vegetation treatments (coverage and removal) for the purpose of exploring the effects of vegetation on the Hg fluxes between the soil and the atmosphere. The soil GEM emissions were continuously measured for a 24-h duration at the center of each plot. Due to the fact that only a single instrument and a one-chamber system was available, the measurements were conducted sequentially from plot to plot during the spring (9 May to 11 May); summer (26 to 28 June; 21 to 23 July; and 22 to 24 August); and fall (27 to 29 September) in 2018. The measurements were obtained using a dynamic flux chamber connected to a RA-915M mercury vapor analyzer. 

Surface soil samples (0–5 cm) were collected and sealed in clean, lucifugal plastic bags for every condition. After being transported to the lab, the samples were air-dried, milled, and sieved to pass through 80 mesh-screen. The Hg concentrations of soil samples were analyzed by UMA (universal mercury attachment) of the Lumex RA-915+, based on the pyrolysis technology to release gaseous Hg^0^ from the sample to the test pool with the detection limit of 0.1 ng·g^−1^.

The gaseous elemental mercury (GEM) was monitored using an automated Hg vapor detector produced by LUMEX RA-915+ (Russia). To be more specific, as a real-time Hg detector, the RA-915+ is based on the Zeeman cold vapor absorption spectrometry technique, with a time resolution of one second. It is calibrated with an internal Hg vapor source. Its real-time detection limit and dynamic detection extents are 2 ng/m^3^ and 5 ng/m^3^ to 2 × 104 ng/m^3^, respectively. The average GEM concentrations were recorded every five minutes in each sampling group, and all of the measurement processes were carried out for a duration of 24 h.

The chamber was placed over the soil surface with the vegetation coverage in order to measure the Hg fluxes, as detailed in Figure 2a,c. Then, the above-ground plants were mowed and removed from the chamber, as shown in Figure 2b. The mowing of the vegetation simulated grassland utilization, such as livestock grazing, particularly those occupied by pastoralists. All the measurements were performed on sunny days only, with leaf-litter present in order to mimic the original environmental conditions. In addition, for the purpose of exploring the relationships between meteorological variables and the soil Hg^0^ evasion, a linear regression analysis method was adopted using the hourly data from each plot across each season. In addition, using the dynamic flux chamber (DFC), the approximate 24-h Hg flux measurements under each treatment were partitioned into four periods as follows: morning (6:00 to 12:00); afternoon (12:00 to 18:00); before midnight (18:00 to 24:00); and after midnight (0:00 to 6:00), respectively. Six distinct sets of measurements were taken per day for each period.

### 2.3. Soil-Air Hg^0^ Flux Measurements

In the present investigation, the Hg^0^ flux values were continuously sampled for 24 h using a cuboid dynamic flux chamber (DFC, 25 cm (length) × 40 cm (width) × 50 cm (height)) [30] coupled to a LUMEX^R^ RA-915+ portable Hg vapor analyzer for each condition. Organic glass was chosen for the chamber because of its transparency and light and low chamber blanks. The chamber was linked through the outlet with the mercury analyzer by a Teflon™ tube (internal diameter of 0.635 cm). The diameter of the chamber inlet was slightly larger than the outer diameter of the Teflon™ tube (outer diameter of 0.8 cm) so as to ensure good airtightness. During measurement, the flux chamber was put on the surface of the area to be measured, the bottom plate of the chamber was extracted, the edge of the chamber penetrated the soil for 2 cm, and a layer of soil around the flux covered chamber to improve the airtightness of the device. Ambient air was pumped throughout the chamber at a constant rate of 0.9 m^3^·h^−1^. The instrument was calibrated using an internal test cell prior to each sampling period [31]. The air intake was positioned at approximately 5 cm above the ground on the middle side (25 cm × 50 cm). The DFC inlet and outlet gases were sequentially sampled by the Hg vapor analyzer at 10-min intervals (two × 5 min samples). The inlet Hg concentrations were analyzed as the ambient Hg^0^ concentrations. The Hg^0^ exchange fluxes of the soil surfaces and the air were calculated according to the following formula:F = (Q·(C_out_ − C_in_))/A(1)
where F represents the flux of the gaseous Hg, consisting mainly of Hg^0^ (ng·m^−2^·h^−1^); A refers to the bottom surface area of the chamber, which was equal to 0.1 m^2^; Q is the flushing flow rate through the chamber, equal to 0.9 m^3^·h^−1^; and C_out_ and C_in_ represent the Hg^0^ concentrations of the DFC outlet and inlet gas samples, respectively (ng·m^−3^). In the present study, in order to prevent the possibility of underestimating the Hg fluxes at low flushing flow rates, and to avoid the underestimation of Hg fluxes due to long DFC turnover times (TOT) [32,33], a constant and relatively high flow rate (0.9 m^3^·h^−1^; TOT = 0.32 min) was applied, which was consistent with the methods applied in previous related studies [13,34,35]. The mean values of two C_out_ and four C_in_ (before and after the C_out_) were used to calculate the Hg fluxes between the soil and air sequentially. One Hg flux datum was obtained every 10 min. The positive and negative results calculated using Equation (1) represented the Hg emissions from the soil and air Hg dry deposition, respectively [36]. A quartz tube filled with dry soda-lime was assembled in front of the instrument gas inlet for the purpose of removing particles and acidic gases. In the field, the instrument was manually calibrated using an internal Hg^0^ permeation source before and after the sampling at each site. In addition, prior to the field campaign, the DFC was cleaned using 10% nitric acid followed by ultrapure water. The DFC blank (0.5 ± 0.3 ng m^−2^·h^−1^; n = 20) was found to be low and not corrected for in the reported flux (Equation (1)). Then, following the Hg^0^ flux sampling, the surface soil layers (approximately 5 cm) were covered in, and the DFC footprint was collected for subsequent total soil mercury (TSM) content analysis in this study’s laboratory facilities.

In addition, parallel to Hg^0^ flux measurements, synchronized 5-min averaged meteorological data were recorded at each sampling site using a portable weather station (ZX-SCQ4, China), including solar radiation, air temperature (inside the chamber), and relative humidity (RH). Furthermore, we used a soil hygrometer to measure the average temperature of 0-5 cm soil and soil moisture (refers to mass of moisture in 100 g dried soil).

### 2.4. Statistical Analysis Results

This study’s statistical analysis was performed using SPSS 26.0 and Origin 9.5. In addition, since the data followed normal distribution patterns, independent-samples *t*-tests were applied in order to analyze the differences in GEM and TSM. In addition, one-way analysis of variance (ANOVA) was used to analyze the variations in the Hg fluxes between the different examined conditions. Pearson’s correlation analysis was applied to analyze the correlations between the Hg fluxes and measured parameters (for example, solar radiation, air temperature, and soil temperature). In the current study, the statistical analysis results were considered significant at a confidence level of 0.95.

## 3. Results

### 3.1. Gaseous Elemental Mercury (GEM) Concentrations in the Ambient Air

The monthly mean daily GEM concentrations in the ambient air above the vegetation coverage/removal and bare soil were 14.9 ± 6.3 to 16.3 ± 6.5; 12.4 ± 3.6 to 16.4 ± 5.6; and 12.4 ± 3.6 to 16.4 ± 5.6 ng·m^−3^, respectively (Table 1), and had ranged between 4.0 and 30.0 ng·m^−3^. The peak value of 16.4 ± 5.6 ng·m^−3^ in and valley value of 12.4 ± 3.6 ng·m^−3^, respectively, occurred during the months of July and August, under the vegetation removal and bare soil conditions, respectively. Significant differences in the GEM concentrations in the ambient air between vegetation coverage and the removal were observed (*p* < 0.05). For example, when the vegetation was removed, the GEM concentrations showed substantial seasonal variations (*p* < 0.05). The GEM concentrations were generally higher than background levels for the northern hemisphere, which is considered to be approximately 1.5 ng·m^−3^.

### 3.2. Total Mercury Concentrations in the Soil

The concentration levels of total Hg in the soil of the study area ranged between 32.8 and 36.2 ng·g^−1^ at a depth range of 0 to 5 cm (Table 2). The total Hg concentrations of the soil ranged between 32.8 ± 1.0 and 36.2 ± 0.4 ng·g^−1^ and peaked in the bare soil in July. On the ground where the vegetation was removed, the total Hg concentrations in soil were observed to have slightly declined when compared with the Hg concentrations with vegetation coverage. Moreover, they were not statistically different (*p* > 0.05). All of the soil Hg concentrations were lower than those previously reported (those observed in urban grasslands between 718 ± 1517 ng·m^−2^·h^−1^ and 4115 ± 1512 ng·m^−2^·h^−1^ [36] and Hg mining areas 33 to 3638 ng·m^−2^·h^−1^) [37]) in soil impacted by human activities and naturally enriched terrestrial landscapes in China.

### 3.3. General Characteristics s of the GEM Fluxes over the Soil-Air Interfaces

As detailed in Table 3, the results revealed that the Hg fluxes ranged between −27.0 and 47.3 ng·m^−2^·h^−1^, with means of 5.4 ± 14.7 to 6.7 ± 14.8; 5.9 ± 14.2 to 7.1 ± 13.6; and 5.9 ± 11.9 to 7.1 ± 13.6 ng·m^−2^·h^−1^ for the *A. anethifolia* coverage, removal, and bare soil, respectively, during the 15-day measurement period (May to September). The lowest and highest Hg fluxes were observed for the *A. anethifolia* coverage in June and without vegetation in July, respectively. The GEM exchange fluxes with vegetation coverage were found to be significantly lower than those in the cases of vegetation removal (*p* < 0.05). Therefore, it is suggested that the vegetation covering the soil played an important role in the Hg exchanges. In total, 2558 of 4320 data points (59%) were net emissions. The ratios of emissions and depositions were found to change seasonally, with increases observed from May to July and decreases from July to September. 

Based on this study’s results, it was determined that vegetation coverage could potentially inhibit Hg emission from the soil. The treatment differences between the vegetation coverage, vegetation removal, and bare soil were all significant in July and August (*p* < 0.05), which was the most thriving period of plant growth (Figure 3). In addition, it was found that the plant community compositions had substantial effects on the Hg fluxes. In summary, during the vegetation growing periods, significant differences in the Hg fluxes were found to exist between the vegetation coverage and the vegetation removal (*p* < 0.05).

Furthermore, there were clear diurnal fluctuations in Hg fluxes for the three treatments, as illustrated in Figure 4. The Hg was generally released from the soil to the air during the daytime hours and deposited from the air to the soil during the nighttime hours, with peaks occurring during the middle of the day at the same time as peaks occurred in the solar radiation levels. Generally speaking, the GEM fluxes began to increase after sunrise sharply. The peaking values were observed at approximately noon (12:00 to 14:00). The valley values were observed at night before sunrise (1:00 to 3:00) and near midnight (23:00 to 24:00). The dawn (4:00 to 6:00) and late evening (18:00 to 20:00) time periods were the turning points between emissions and depositions. In nearly all cases, the peak Hg fluxes from the soil surfaces with vegetation coverage were lower than those with vegetation removed.

### 3.4. Environmental Factors Influencing the Mercury Soil-Air Exchanges

In the present investigation, in order to clarify the mechanism of Hg emissions for the three treatments, linear regression and Spearman correlation analyses between the Hg flux values and the meteorological parameters were derived (Figure 5, Table 4). The results revealed that higher Hg emission levels were observed when the solar radiation and air/soil temperature levels were high under all the examined conditions. In addition, negative correlations between the air/soil humidity and the Hg fluxes were observed in this study. However, it was determined that solar radiation was the most important factor influencing the Hg exchange fluxes under all the examined conditions (R^2^ = 0.4337 to 0.5307). The second most important factor was air temperature (R^2^ = 0.2548 to 0.3561). Those findings were in agreement with the results of previous studies [32]. Generally speaking, the increased humidity corresponded well with the decreasing solar radiation and temperature levels. However, a positive correlation between the soil moisture and the Hg fluxes has been noted in previous studies [38,39].

## 4. Discussion

### 4.1. General Characteristics of the GEM Fluxes

#### 4.1.1. Characteristics of the GEM Fluxes of Different Terrestrial Surfaces 

This research investigation presented the soil-air Hg exchange flux measurements in a degraded meadow steppe located in northeastern China. Similar studies regarding GEM fluxes over other terrestrial surfaces have been conducted. (Table 5). The data used below is based on the DFC method. The Hg fluxes in the current study (5.4 ± 14.7 to 6.7 ± 14.8; 5.9 ± 14.2 to 7.1 ± 13.6; and 5.9 ± 11.9 to 7.1 ± 13.6 ng·m^−2^·h^−1^) were observed to be much higher than the global natural emission (0.7 to 1.1 ng·m^−2^·h^−1^) [40]. However, the results were comparable to those observed in the studies conducted in the Changbai temperate forests (4.4 ± 28.74 ng·m^−2^·h^−1^) [41] and croplands (−11.8 to 7.1 ng·m^−2^·h^−1^) in northeastern China [42]. Moreover, when compared with Hg polluted areas, the fluxes were remarkably lower than those observed in urban grasslands (between 718 ± 1517 ng·m^−2^·h^−1^ and 4115 ± 1512 ng·m^−2^·h^−1^) [36] and Hg mining areas (33 to 3638 ng·m^−2^·h^−1^) [37]. This study determined that the important reasons for the higher Hg fluxes in the aforementioned areas were the impacts of human production and livelihood pursuits, such as coal consumption and metallurgy, which contributed to elevated GEM concentrations. In contrast, the Hg emissions from soil in natural meadow steppe areas are the results of automatic processes which are only minimally affected by pollution or human activities.

#### 4.1.2. Diel Variations in the Gaseous Elemental Mercury Concentration Levels

In the present study, there were clear diurnal fluctuations in the Hg fluxes under the three conditions, as illustrated in Figure 3. Generally speaking, Hg was released from the soil to the air during the daytime and deposited from the air to the soil during the nighttime, with peaks occurring during the middle of the day at the same time as the peaks in solar radiation. This was one of the reasons why solar radiation was considered to be the most important environmental parameter affecting the emissions of soil mercury to the atmosphere. That conclusion agreed well with the findings of previous studies [29,43]. The turning points of the emission and deposition processes were approximately 6:00 (morning) and 18:00 (evening). It was observed that in nearly all cases, the peaks of the Hg fluxes from the soil surfaces with vegetation coverage were lower than those with vegetation removed.

#### 4.1.3. Seasonal Variations in the Gaseous Elemental Mercury Concentration Levels

The gaseous elemental mercury (GEM) concentrations in the ambient air presented seasonal variations during the observation periods (Figure 6). These variations were apparent, with the mean GEM concentrations decreasing with the order of August > June > May > September > July. The differences in the GEM during all the months were found to be statistically significant (*p* < 0.05), with the exceptions of June and August. The GEM concentrations in July were the lowest, while the Hg exchange fluxes were the highest. This may have been due to the influencing effects of the growing plants absorbing the gaseous Hg through their stoma [44]. Plants are known to accumulate Hg in their above-ground biomasses over the growing season [45]. In the present investigation, it was clear that the GEM concentrations were significantly elevated when compared with the background atmospheric Hg concentrations in China (1.7 ng·m^−3^ in summer and 0.6 ng·m^−3^ in winter) [30].

### 4.2. Impacts of Vegetation Coverage on Hg Fluxes

In the present study, it was observed that the GEM fluxes with vegetation coverage were significantly lower than those under the conditions of removed vegetation. These findings indicated that plants may potentially inhibit Hg emissions from soil layers (Figure 2). Vegetation is known to have considerable impacts on air/soil Hg fluxes through the following two pathways: 1. Altering environmental variables at the soil-air interfaces (for example, reducing solar radiation and temperature levels and changing soil properties) [46]; 2. Direct absorption of Hg by foliage [32]. However, when the plants were removed, the environmental variables had not displayed significant differences between vegetation coverage and vegetation removal in this study (independent-sample *t*-test; *p* > 0.05). The increased Hg deposition events observed in this study’s experiments were interesting phenomena that suggested that plants in the natural meadow steppe may have inhibited the Hg emissions from the soil mainly through absorbing atmospheric Hg. The net ecosystem flux is the balance between soil emissions and vegetation uptake, which provides an overall estimate of whether a region is a net source or a net sink of Hg to the atmosphere. According to the results obtained in this study, the examined meadow steppe region was a terrestrial source of the regional Hg budget. With regard to the influencing effects of the grass coverage, it was observed that the contribution was lower than that without vegetation. Those results corresponded with a review of recent Hg flux data obtained in the northwestern American states [46]. Therefore, when accounting for the uptake of gaseous Hg by vegetation, the results had shifted to 62% of the terrestrial surfaces having net emissions [38].

The effects of vegetation on Hg fluxes have also been observed in the forest canopy studies conducted by Mazur et al. [47]. However, the influencing mechanisms in forested areas are not entirely the same as those in meadow steppe areas. The harvesting processes in forested lands tend to lead to wetter soil, with substantially more solar radiation reaching the forest floor. Studies have identified significant differences in Hg fluxes based on the different types of forest harvesting (clear-cut + biomass, clear-cutting, and controlled). Since biomass harvesting removes a larger quantity of vegetation when compared with traditional clear-cutting, the soil surface layers become exposed to proportionately more solar radiation. In addition, canopy shading through the growing seasons effectively limits the magnitude of Hg emissions from lower levels when compared with those observed under canopies with minimal shading, regardless of the higher air and soil temperatures experienced during those periods [39,48].

### 4.3. Impacts of the Meteorological Conditions and the Air/Soil Hg Content Levels

The measurements of GEM fluxes under different coverage and meteorological conditions allowed the relationships between fluxes and meteorological conditions to be investigated. Figure 5 details the correlation coefficients for the GEM fluxes against the meteorological parameters under specific conditions. It can be seen in the figure that there were strong correlations between the fluxes and five meteorological parameters under the different coverage conditions. In summary, positive correlations were observed between the fluxes and the solar radiation and between the fluxes and the air/soil temperatures. However, negative correlations were observed between the fluxes and the relative humidity levels and between the fluxes and the soil moisture values.

#### 4.3.1. Solar Radiation Levels

This study’s comparison results of the correlation coefficients of the five meteorological parameters revealed that the highest positive correlation coefficients were between the fluxes and the solar radiation. Therefore, solar radiation was considered to be the most important factor affecting the GEM fluxes. Solar radiation is widely recognized as one of the most important environmental parameters affecting Hg emissions from the soil to the atmosphere. Solar radiation promotes the photo-reduction of Hg^2+^ to Hg^0^ [49]. During the month of July in this study, when the solar radiation was at the maximum level during the entire growing season, the Hg^2+^ photo-reduction had proceeded at the highest rate. This was similar to the results of a study conducted in Australian alpine grasslands by Howard et al. [26]. In addition, it has been determined that intense solar radiation levels will lead to higher air and soil temperatures, which tend to accelerate the biotic or abiotic transformation of Hg^0^ and enhance the mercury vapor pressure to facilitate the volatilization of mercury [50]. 

In the current investigation, higher correlation coefficients were obtained between the flux and solar radiation values under the *A**. anethifolia* removal condition when compared with the *A**. anethifolia* cover condition. These findings suggest that meteorological conditions have weaker effects on fluxes under soil-covered conditions. Furthermore, some studies have shown that the thermal energy absorbed by plants from solar radiation can increase the mercury vapor pressure and transport mercury from plants into the atmosphere by convection [47]. However, that type of phenomenon was not obvious in the current study. Therefore, it was concluded that the correlation between the solar radiation and mercury flux had increased with the decrease in vegetation coverage.

#### 4.3.2. Air and Soil Temperature Values

Significant positive relationships (*p* < 0.01) between Hg fluxes and the air/soil temperature values were observed under all the examined conditions in this study, which was similar to the results obtained by Tao et al. [32], Choi and Holsen [51], and Lindberg [52]. The coefficients between the fluxes and the air temperature values were second only to that of the solar radiation. During the measurement processes, it was determined that the Hg^2+^ photo-reduction was driven by solar radiation rather than by air temperature. Therefore, solar radiation was concluded to be the main factor affecting the fluxes. The increases in solar radiation not only accelerated the photo-reduction but also increased the air temperature levels, thereby promoting Hg emissions [47]. The influences of air temperature on the Hg emissions mainly involved increasing the vapor pressure and activation energy of the chemical reactions. That mechanism is typically described using the Arrhenius Equation [47]. 

It has been proposed that there are at least two sources of Hg^0^ emitted from soil: a natural “pool” of Hg^0,^ which is primarily absorbed on the surface, and Hg^2+^, which can be photochemically reduced to Hg^0^ by sunlight [23]. Gustin suggested that elevated soil temperature levels can accelerate Hg desorption from soil and its movement up through soil columns [53]. Therefore, under dark conditions, the primary factors leading to emissions of Hg from soil are generally considered to be related to the enthalpy of volatilization. However, under light conditions, the emissions will be related to both the enthalpy of volatilization and the photo-reduction process. In summary, solar radiation drives Hg emissions during daylight hours while soil temperature levels become the most important driving factor during nighttime hours.

#### 4.3.3. Air Humidity and Soil Moisture Values 

In the present investigation, the GEM fluxes and air relative humidity/soil moisture values were found to be negatively correlated under the three examined conditions. However, based on the currently obtained results, no clear conclusion was reached regarding the effects of relative humidity on Hg fluxes. For example, some studies have found negative correlations between Hg fluxes and relative humidity levels [43,51]. Meanwhile, other studies have observed positive correlations [38] or even no correlations [54]. In this study, it was observed that the Hg fluxes were negatively correlated with the relative humidity. Generally speaking, increases in solar radiation and air temperature tend to lead to decreases in relative humidity, which may be the main cause of the negative correlation. Second, increased relative humidity can potentially combine vapor and mercury. Meanwhile, vapor can occupy air gaps in the soil and block the release of soil gases [55].

In this study, soil moisture was found to be negatively correlated with the fluxes. This had differed from several previous reported results, in which positive correlations between the soil moisture and the Hg fluxes were identified under controlled laboratory settings, as well as during some field investigations [55,56]. Precipitation has also been believed to influence Hg emissions in several ways, such as the desorption of Hg^0^ in the soil matrix by water molecules; reduction in Hg^2+^ by dissolved organic matter, O_3_, Fe^2+^, and/or biotic mechanisms, and then transported to the surface with water vapor [55]. However, it was determined that one of the most important reasons for the differing results was that there were no rainy days during this study’s measurement processes, and the soil moisture varied diurnally followed by high solar radiation levels. 

#### 4.3.4. GEM Concentrations in the Air and Soil

Air is an important Hg “pool” and can influence the Hg transmission to other “pools.” Recent studies have shown the ambient-air Hg^0^ concentration levels determine the direction of Hg movement [5]. If the Hg^0^ concentrations are at relatively low levels (3.26 to 10.8 ng·m^−3^), the fluxes will be positively related to the Hg^0^ concentrations [57]. However, the studies conducted in high-level Hg regions have indicated that high air Hg^0^ concentrations (122 to 284 ng·m^−3^) have negative effects on released fluxes and may even result in air Hg^0^ sediment in the soil, although the soil will contain more Hg (150 to 260 ng·g^−1^) than local background level [57]. In this study, the air Hg^0^ concentrations were observed to be positively correlated with the Hg fluxes in all the obtained measurements (*p* < 0.05). The possible reason was that the air Hg^0^ concentrations were driven by the emission fluxes. It was speculated that this had also led to the observed air Hg^0^ diel and seasonal variable patterns.

Along with the air pools, the pool effects of the soil are also considered to be very important. Numerous studies have reported that there is a significantly positive correlation between the soil Hg concentrations and the surface-Hg fluxes [5]. In this study, not only no significant correlations could be found between the soil Hg concentrations and the fluxes but also there were unobvious seasonal varieties in the soil Hg concentrations, with the maximum values reached in July. A common explanation is that the temperature and rainfall levels in July had promoted organic matter accumulation, which led to the strong sorption of Hg to functional groups on the soil organic matter [58,59].

## 5. Conclusions

This study represents the first-time measurements of the exchange fluxes of GEM over soil-air interfaces under the conditions of different vegetation types. The study area was located in the Songnen Grasslands region of northeastern China. The results indicated that the GEM fluxes over the soil-air surfaces ranged from -18.0 ng·m^−2^·h^−1^ to 47.3 ng·m^−2^·h^−1^, with a mean value of 4.3 ng·m^−2^·h^−1^ to 9.0 ng·m^−2^·h^−1^ at the plant coverage and removal treatments in the Songnen Grasslands, during the growing season in 2018. Therefore, it was assumed that natural meadow steppe regions are significant local atmospheric emission sources. In this study, the fluxes exhibited diurnal and seasonal patterns, in which the emissions were usually observed during the daytime hours, the depositions were observed during the nighttime hours, and the peak flux values were reached during the summer (July). This study also found that vegetation played an important role in the GEM exchange processes in that it inhibited the emission fluxes through absorbing atmospheric Hg. Moreover, the fluxes were found to have significant correlations with solar radiation, air/soil temperature, and air/soil humidity levels. However, solar radiation was determined to be the most important factor affecting the Hg fluxes in the Songnen Grasslands steppe region. Overall, the data obtained in this study provided a basis for estimating the Hg budget from similar terrestrial surfaces and also supported the prior findings regarding the relationships between environmental factors and Hg fluxes.

## Figures and Tables

**Figure 1 biology-10-00917-f001:**
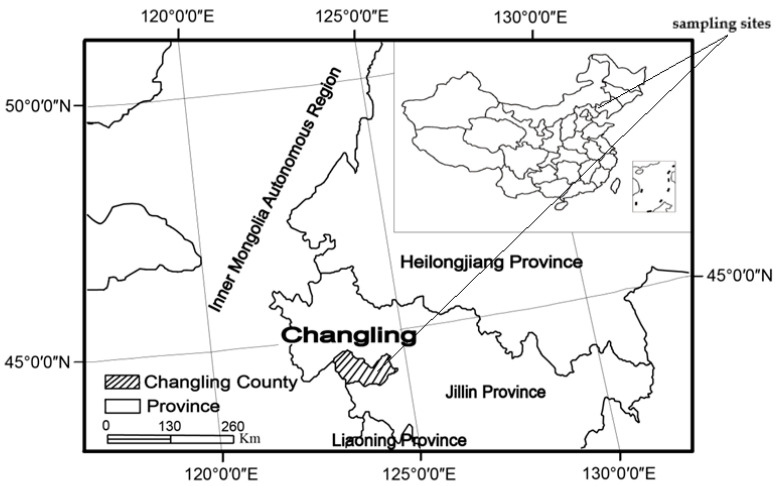
Map of the locations of the sampling sites.

**Figure 2 biology-10-00917-f002:**
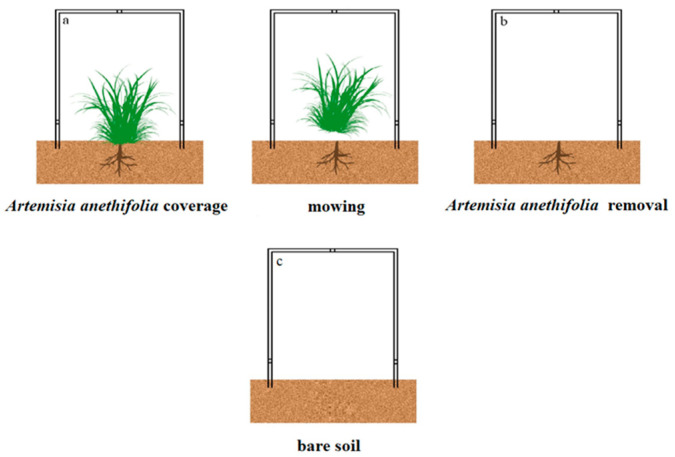
Schematic diagram of the experimental design: (**a**,**b**) *Artemisia anethifolia* coverage and removal; and (**c**) Bare soil.

**Figure 3 biology-10-00917-f003:**
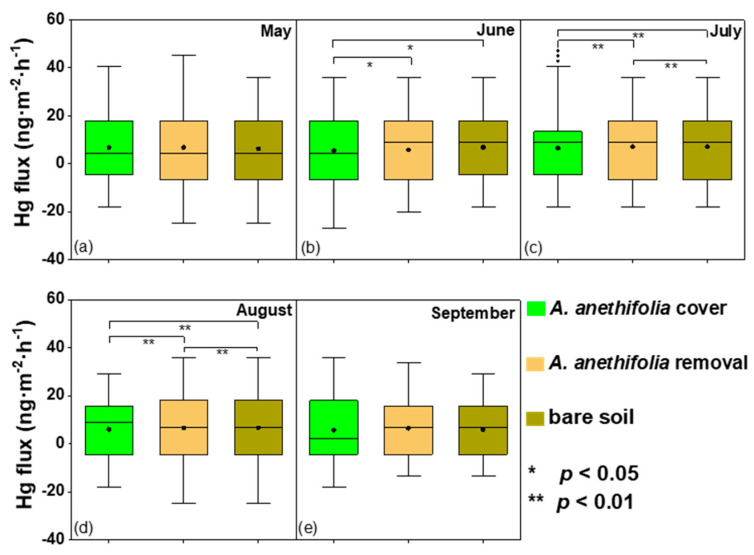
Box plots of the soil-air Hg exchange fluxes for the three examined conditions (*Artemisia anethifolia* coverage, *Artemisia anethifolia* removal, and bare soil) from May to September 2018. *: In the figure, the asterisk represents a significantly different flux compared to the other plots during the same season; sub figure (**a**–**e**) represents May to September 2018; * *p* < 0.05; ** *p* < 0.01.

**Figure 4 biology-10-00917-f004:**
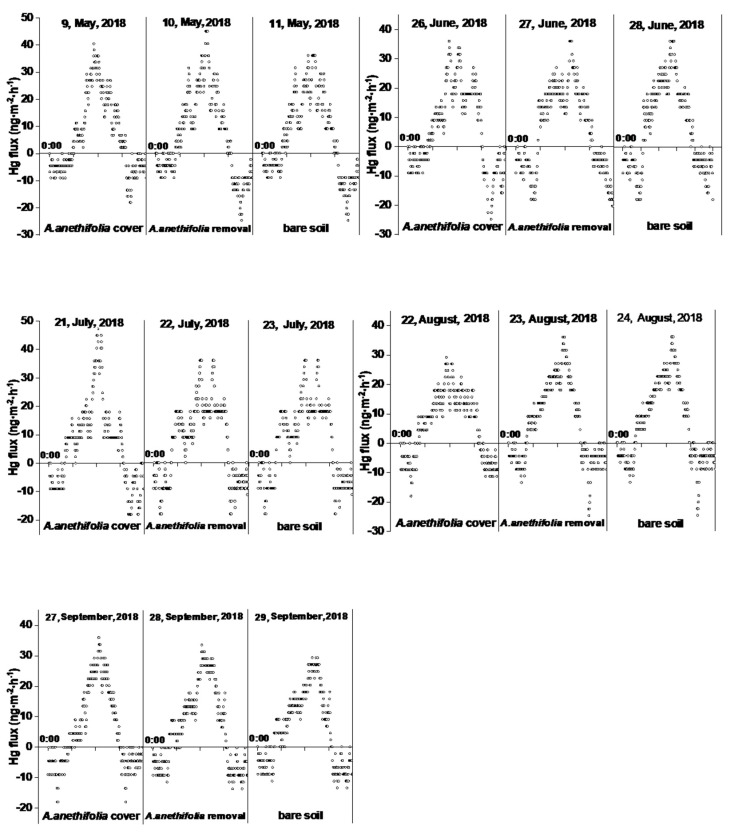
Diurnal variations of the Hg^0^ flux (ng·m^−2^·h^−1^) measurement data for the three conditions during each month.

**Figure 5 biology-10-00917-f005:**
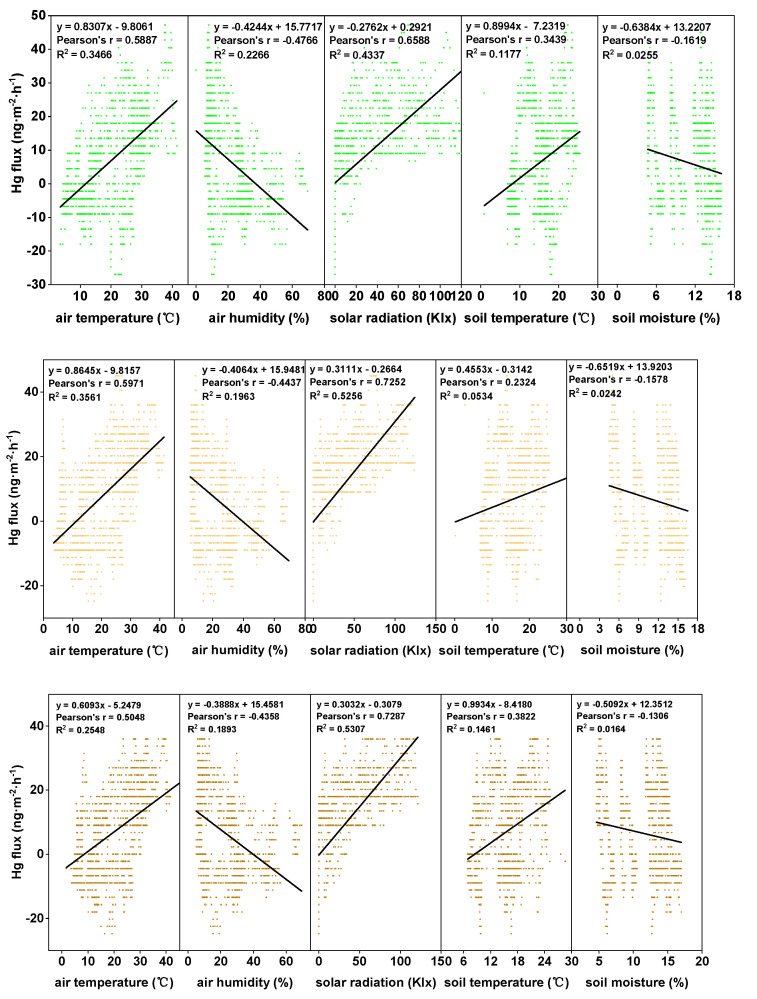
Correlation between the diurnal GEM fluxes with meteorological factors, such as air/soil temperatures, solar radiation, and air/soil humidity levels for the three examined conditions during the entire growing season from top to bottom: Anethifolia cover; Anethifolia removal; and bare soil.

**Figure 6 biology-10-00917-f006:**
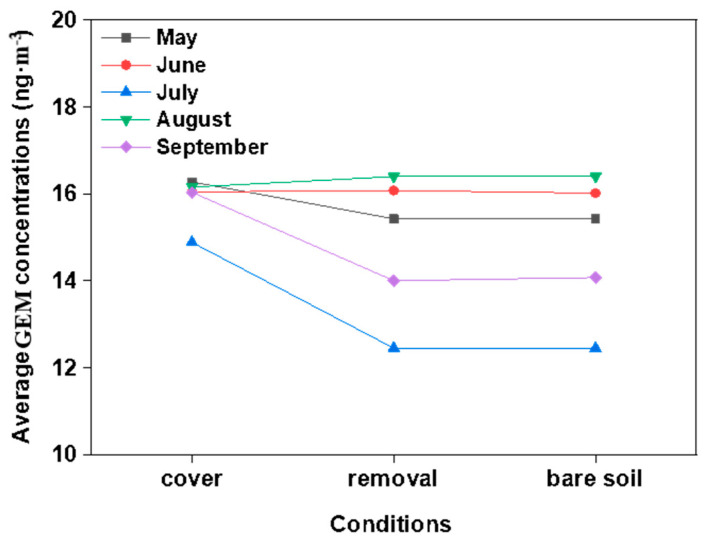
Twenty-four-hour average GEM concentrations under the three examined conditions during the monitored seasons.

**Table 1 biology-10-00917-t001:** Summary of gaseous elemental mercury concentrations (ng·m^−3^) for the diel in 2018 (n = 288).

Treatments	Date	Range	Diel Mean ± SD
*A. anethifolia* cover	9 May	4.0~27.0	16.3 ± 6.5
*A. anethifolia* removal	10 May	5.0~26.0	15.4 ± 6.3
Bare soil	11 May	4.0~26.0	15.4 ± 6.3
*A. anethifolia* cover	26 June	5.0~25.0	16.0 ± 6.0
*A. anethifolia* removal	27 June	7.0~26.0	16.1 ± 6.3
Bare soil	28 June	7.0~25.0	16.0 ± 6.2
*A. anethifolia* cover	21 July	7.0~26.0	14.9 ± 6.3
*A. anethifolia* removal	22 July	6.0~19.0	12.4 ± 3.6
Bare soil	23 July	6.0~19.0	12.4 ± 3.6
*A. anethifolia* cover	22 August	8.0~26.0	16.2 ± 5.8
*A. anethifolia* removal	23 August	6.0~26.0	16.4 ± 5.6
Bare soil	24 August	6.0~26.0	16.4 ± 5.6
*A. anethifolia* cover	27 September	7.0~30.0	16.0 ± 6.8
*A. anethifolia* removal	28 September	6.0~27.0	14.0 ± 6.5
Bare soil	29 September	6.0~28.0	14.1 ± 6.6

**Table 2 biology-10-00917-t002:** Total mercury concentrations (mean ± SD) of the soil (ng·g^−1^) for the three treatments in 2018.

Treatments	Month				
	5	6	7	8	9
*A. anethifolia* cover	34.8 ± 1.5	33.2 ± 1.0	35.4 ± 1.8	33.8 ± 0.7	33.2 ± 0.7
*A. anethifolia* removal	32.8 ± 0.8	32.8 ± 0.7	35.4 ± 0.5	33.8 ± 1.0	32.8 ± 1.0
Bare soil	35.2 ± 1.0	35.6 ± 0.5	36.2 ± 0.4	34.4 ± 0.8	33.4 ± 0.4

**Table 3 biology-10-00917-t003:** Soil-air Hg exchange fluxes (ng·m^−2^·h^−1^) for diel and the separate released and deposited periods based on measurements during the three treatments (n = 288, n = n_1_ + n_2_).

Treatments	Date	Mean Hg Flux (Range)	SD	Emission	SD	n_1_	Deposition	SD	n_2_
*A. anethifolia* cover	9 May	6.7 (−18~40.5)	13.8	17.0	10.4	157	−6.0	3.7	131
*A. anethifolia* removal	10 May	6.7 (−24.8~45)	15.6	18.1	11.0	162	−8.6	4.9	126
Bare soil	11 May	6.2 (−24.8~36)	14.7	17.2	9.7	162	−8.6	4.9	126
*A. anethifolia* cover	26 June	5.4 (−27~36)	14.7	14.5	10.2	188	−10.3	7.1	100
*A. anethifolia* removal	27 June	5.9 (−20.0~36.0)	14.2	13.4	12.1	184	−7.5	4.7.	104
Bare soil	28 June	6.9 (−18.0~36.0)	14.3	14.2	12.8	184	−6.1	3.3	104
*A. anethifolia* cover	21 July	6.5 (−18~47.3)	14.5	15.5	10.8	176	−9.2	4.5	112
*A. anethifolia* removal	22 July	7.1(−18.0-36.0)	13.6	17.2	7.2	170	−8.5	3.6	118
Bare soil	23 July	7.1 (−18.0~36)	13.6	17.2	7.2	170	−8.5	3.6	118
*A. anethifolia* cover	22 August	6.0 (−18.0~29.0)	11.0	13.7	6.0	177	−7.3	3.0	111
*A. anethifolia* removal	23 August	6.6 (−24.8~36)	13.1	14.8	9..8	176	−7.1	4.3	112
Bare soil	24 August	6.6 (−24.8~36.0)	13.1	14.8	9.8	176	−7.1	4.3	112
*A. anethifolia* cover	27 September	5.7 (−18.0~36.0)	12.7	15.9	9.5	148	−6.3	3.5	140
*A. anethifolia* removal	28 September	6.5 (−13.5~33.8)	12.8	16.1	8.4	164	−7.2	2.7	124
Bare soil	29 September	5.9 (−13.5~29.3)	11.9	14.9	7.2	164	−7.2	2.7	124

**Table 4 biology-10-00917-t004:** The Pearson’s correlation between Hg fluxes and environmental variables under three treatments.

Treatments	Air Temperature	Relative Humidity	Solar Radiation	Soil Temperature	Soil Moisture
forbs coverage	*p* = 0.001	*p* = 0.000	*p =* 0.000	*p* = 0.021	*p* = 0.032
forbs removal	*p* = 0.001	*p* = 0.001	*p* = 0.000	*p* = 0.023	*p* = 0.034
Bare soil	*p* = 0.001	*p*= 0.001	*p* = 0.000	*p* = 0.015	*p* = 0.036

**Table 5 biology-10-00917-t005:** Characteristics of the Hg fluxes of different terrestrial surfaces in northeastern China.

Terrestrial Surfaces	Hg Fluxes (ng·m^−2^·h^−1^)
Global natural emissions	0.7–1.1
*A. anethifolia* cover	5.4 ± 14.7–6.7 ± 14.8
*A. anethifolia* removal	5.9 ± 14.2–7.1 ± 13.6
Bare soil	5.9 ± 11.9–7.1 ± 13.6
Changbai temperate forest	4.4 ± 28.74
Changbai temperate cropland	−11.8–7.1
Urban grassland	718 ± 1517 and 4115 ± 1512
Hg-mining area	33–3638

## Data Availability

The data presented in this study are available on request from the corresponding author.

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
