# Peer review of "Gaseous Elemental Mercury Exchange Fluxes over Air-Soil Interfaces in the Degraded Grasslands of Northeastern China"

_biology, 2021, doi:10.3390/biology10090917_

Round 1
Reviewer 1 Report
Review on Zhang et al
The authors describe a measurement of GEM fluxes over in the steppe in China. Such measurements are needed. However, the problem with this paper is that the experimental methods are not properly described to allow a judgment about the significance of the results. The confusion of terminology and units shows authors´ deficits in understanding of measurement techniques and atmospheric chemistry. Consequently, I recommend the publication of the paper after an extensive modification, if at all.
Line 29: Lumex can measure only elemental mercury, not an oxidized one. TGM stands for “total gaseous mercury” and, as such, cannot be used for Lumex measurements.
Line 34: The daily mean ….
Line 35: “ambient levels of TGM” in units of ng g-1?
Line 43: “air humidity” – absolute or relative?
Figure 1: It is not clear in which part of China the measurements were done.
Section 2: How was Hg in soil determined? Is the soil Hg content related to dry or wet weight? How was soil temperature measured – in which depth?
Line 153: Lumex can measure only gaseous elemental mercury (GEM). Please correct throughout the text.
Figure 2: What is the difference between “bare soil” and soil where the plants were mowed away?
Fig 6 shows hardly any difference?
Line 179: The dimensions of the chamber need to be specified: e.g. what is the height? What was the material of which the chamber was made?
Line 181-182: What are the dimensions of the inlet? What was the position of the outlet? How was the gap between soil and the chamber sealed? How can you be sure that the air flows through the inlet and not through the gap between the chamber and the soil? The air flow resistance of the soil is surprisingly small. If the inlet is not large enough you suck the air through the soil. Have you tested this?
Lines 193-194: The optimum flow rate through the chamber has to be established experimentally
because it can vary from site to site. To rely on literature data is thus not sufficient.
Lines 194 and 190: The flow rates are inconsistent.
Lines 209-212: Air temperature inside or outside of the chamber? Soil temperature – at which depth? How is the soil moisture defined – percent of what?
Line 218: Pearson
Line 224-225: “above vegetation” and “above bare soil”?
Line 227: “valley”? You mean “minimum”. Please correct throughout the text.
Table 2: Soil content related to dry or wet soil weight?
Figure 3: The figure caption should contain information about the three examined conditions.
Paragraph 291-303: Because the air humidity is given in % you mean relative humidity, not the
absolute one. The problem with relative humidity is that it is inversely related to temperature at constant absolute one. As such relative humidity is redundant to air temperature.
Section 4.1.1: DFC methods are generally not suitable to measure the real fluxes because the
measured fluxes are influenced by the chamber: e.g. the temperature in the chamber is substantially higher than outside of it. DFC methods are quite useful to study the flux mechanisms but not to determine the real fluxes. Comparison with other sites in Table 4 where possibly the real fluxes were measured by gradient or micrometeorological methods is thus misleading.
Section 4.1.3 and Figure 6: The text seems to be related to air concentration, the figure with units of ng g-1 to soil concentrations. What is correct?
Lines 401-402: As already mentioned, relative air humidity in inversely related to air temperature. As such, negative correlation of RH provides nearly the same information as air temperature and is thus redundant.
Author Response
Major points: (“C” means Comment, “R” means Response)
C1: Line 29: Lumex can measure only elemental mercury, not an oxidized one. TGM stands for “total gaseous mercury” and, as such, cannot be used for Lumex measurements.
R1: We are indeed measuring elemental mercury, so we changed the “TGM” of the full text to “GEM”.
C2: Line 34: The daily mean ….
R2: We replaced: “The mean diel” with “The daily mean” to make the expression clearer. Line 34, Page 2.
C3: Line 35: “ambient levels of TGM” in units of ng g-1?
R3: We changed ng. g-1 to ng.m-3. Line 36, Page 1.
C4: Line 43: “air humidity” – absolute or relative?
R4: We added “relative” between “air” and “humidity”. Line 43, Page 1.
C5: Figure 1: It is not clear in which part of China the measurements were done.
R5: We changed the description of the study area to correspond to the Figure 1. Please see Line 123-124, Page 3.
C6: Section 2: How was Hg in soil determined? Is the soil Hg content related to dry or wet weight? How was soil temperature measured – in which depth?
R6: We have supplemented the measurement method of soil mercury. And we use a soil hygrometer to measure the average temperature of 0-5cm soil. Please see Lines 159-162, Page 4.
C7: Line 153: Lumex can measure only gaseous elemental mercury (GEM). Please correct throughout the text.
R7: We have corrected the “total gaseous mercury (TGM)” of the full text to “gaseous elemental mercury (GEM)”.
C8: Figure 2: What is the difference between “bare soil” and soil where the plants were mowed away?
R8: As you can see in Figure 2(b), we just mowed the above-ground plants and kept the under-ground plants. Thus, the difference between “the plants were mowed away” and “bare soil” is that the former has roots and the latter does not.
C9: Fig 6 shows hardly any difference?
R9: The gaseous elemental mercury (GEM) concentrations in the ambient air presented seasonal variations during the observation periods. These variations were apparent, with the mean GEM concentrations decreasing with the order of August > June > May > September > July. I guess what you mean is that there is no significant difference between the three conditions in June and August, but this is the case.
C10: Line 179: The dimensions of the chamber need to be specified: e.g. what is the height? What was the material of which the chamber was made?
R10: We specified the dimensions of the chamber, (DFC, 25 cm (length) × 40 cm (width) ×50 cm (height)). We have added the description of the chamber material (organic glass) in Lines 193-196, Page 5.
C11: Line 181-182: What are the dimensions of the inlet? What was the position of the outlet? How was the gap between soil and the chamber sealed? How can you be sure that the air flows through the inlet and not through the gap between the chamber and the soil? The air flow resistance of the soil is surprisingly small. If the inlet is not large enough you suck the air through the soil. Have you tested this?
R11: Organic glass was chosen for the chamber because of its transparency and to light and low chamber blanks. The chamber was linked through the outlet with the mercury analyzer by Teflon™ tube (internal diameter of 0.635 cm). Ambient air was pumped throughout the chamber at a constant rate of 0.9 m3·h-1.
C12: Lines 193-194: The optimum flow rate through the chamber has to be established experimentally because it can vary from site to site. To rely on literature data is thus not sufficient.
R12: We used the same method to study the characteristics of grassland mercury pool with low degradation degree in the same area, and always used the flow rate of 0.9. In order to compare and analyze with our previous experiments, we adopted the same flow rate.
C13: Lines 194 and 190: The flow rates are inconsistent.
R13: We corrected the flow rate on line 210 to make it consistent with line 206.
C14: Lines 209-212: Air temperature inside or outside of the chamber? Soil temperature – at which depth? How is the soil moisture defined – percent of what?
R14: Air temperature inside the chamber. We use a soil hygrometer to measure the average temperature of 0-5cm soil and soil moisture (refers to mass of moisture in 100g dried soil). We added relevant information to make the content complete. Please see Lines 225-229, Page 6.
C15: Line 218: Pearson
R15: We corrected it. Line 235, Page 6.
C16: Line 224-225: “above vegetation” and “above bare soil”?
R16: Yes, we revised the inaccurate description in Line 240, Page 6.
C17: Line 227: “valley”? You mean “minimum”. Please correct throughout the text.
R17: In this sentence, we use “peak value” to express “maximum”, here we use “valley value” to express “minimum”, we think this expression is more vivid.
C18: Table 2: Soil content related to dry or wet soil weight?
R18: Dry soil weight. Regarding the determination of soil mercury, we mentioned it on Line 159-164, Page 4. Surface soil samples (0–5 cm) were collected and sealed in clean, lucifugal plastic bags for every condition. After being transported to the lab, the samples were air-dried, milled and sieved to pass through 80 mesh-screen.
C19: Figure 3: The figure caption should contain information about the three examined conditions.
R19: We added information about the three examined conditions in figure caption. Lines 289-290, Page 8.
C20: Paragraph 291-303: Because the air humidity is given in % you mean relative humidity, not the absolute one. The problem with relative humidity is that it is inversely related to temperature at constant absolute one. As such relative humidity is redundant to air temperature.
R20: “Relative humidity is that it is inversely related to temperature at constant absolute one,” is correct, we discussed it to show that we are consistent with previous research results.
C21: Section 4.1.1: DFC methods are generally not suitable to measure the real fluxes because the measured fluxes are influenced by the chamber: e.g. the temperature in the chamber is substantially higher than outside of it. DFC methods are quite useful to study the flux mechanisms but not to determine the real fluxes. Comparison with other sites in Table 4 where possibly the real fluxes were measured by gradient or micrometeorological methods is thus misleading.
R21: The limitations you mentioned about this method exist objectively, and we do not deny other methods such as micro meteorological (MM) approaches. Each method presents its own benefits and drawbacks related to application, with the DFC being portable, simple to deploy, and not subject to the strict site constraints of the MM meteorological, but potentially influencing the system under measurement (e.g., side temperature may be quite different from outside chamber temperature). However, in the past 30 years, many similar studies at home and abroad still use this method, mainly because it is economical and durable and can effectively reduce the long-term experimental cost. In addition, our previous research also used this method. Continue to use this method so that our previous data can be compared. Of course, the data used in Table 4 is based on DFC method.
C22: Section 4.1.3 and Figure 6: The text seems to be related to air concentration, the figure with units of ng g-1 to soil concentrations. What is correct?
R22: The text is correct, we changed the unit “ng. g-1” to “ng.m-3” in Figure 6.
C23: Lines 401-402: As already mentioned, relative air humidity in inversely related to air temperature. As such, negative correlation of RH provides nearly the same information as air temperature and is thus redundant.
R23: Of course, negative correlation of RH provides nearly the same information as air temperature. But we don't think it's redundant. There will be some differences in the relationship between humidity and temperature on grasslands with different humidity types. We can't conclude that the discussion on humidity is meaningless because the negative correlation between humidity and temperature is just reflected in this experiment.
Reviewer 2 Report
In this study, the authors investigated the total gaseous mercury (TGM) exchange fluxes over two land cover types (including Artemisia anethifolia coverage and removal and bare soil) using a dynamic flux chamber. They reported that the grassland soil serves as both a source and a sink for atmospheric Hg, depending on the season and meteorological factors, and that the plants play an important inhibiting role in the Hg exchanges between the soil and the atmosphere. I think that the results presented here are valuable, because similar studies are still scarce in grassland regions. In this manuscript, however, there are ambiguous or inadequate points. Hence, major revision of the manuscript is required to eliminate them. My comments are given below.
Line 35: Revise “ng g-1” to “ng m-3.
Figure 1: The study area should be displayed on the map of China in the upper right of the figure.
Lines 131–132: The authors should explain why A. anethifolia coverage was selected.
Line 179: What is the material of DFC?
Lines 224–226: These values correspond to the daily average TGM concentrations in each month. Clarify this.
Lines 240–242: This view is incorrect, because the differences are not statistically significant as described on line 242.
Lines 242–244: Show the data.
Lines 249–268: Table 3 shows that there are markedly large variations in the measured Hg fluxes. I think that comparing the mean values of such data between the three examined conditions is meaningless even if the differences are statistically significant. Moreover, what are the variations in the Hg fluxes under each condition? Thus, whether significant differences in the Hg fluxes exist truly between the vegetation coverage and the vegetation removal is not clear.
Figure 5: Indicate p values for these correlations. Are these correlations statistically significant?
Line 313: I do not understand this heading.
Line 315: Eliminate “however”.
Line 320: This mean value is meaningless, because the SD is markedly large.
Table 4: Show the references of these data.
Figure 6: Revise “ng g-1” to “ng m-3.
Line 377: Cite the reference.
Lines 490–491: This is incorrect, because the seasonal variations in the soil Hg concentrations are very small (see Table 2).
Line 500: Eliminate “respectively”.
Author Response
Major points: (“C” means Comment, “R” means Response)
C1: Line 35: Revise “ng g-1” to “ng m-3.
R1: Done. Line 36, Page 1.
C2: Figure 1: The study area should be displayed on the map of China in the upper right of the figure.
R2: We have redrawn Figure 1 to meet your requirements.
C3: Lines 131–132: The authors should explain why A. anethifolia coverage was selected.
R3: We added a statement about why we chose A. anethifolia. Artemisia anethifolia is the most salinity-tolerant Artemisia plant, which forms small communities on meadow grassland and dry grassland. The increase of this artemisia is often a sign of overgrazing or grassland degradation. In addition, the growing season of a is from May to September each year, which is the golden period for field experiments in Northeast China. So, we selected Artemisia anethifolia as our experimental species. Lines 132-137, Page 3.
C4: Line 179: What is the material of DFC?
R4: Organic glass was chosen for the chamber because of its transparency and to light and low chamber blanks. Please see Lines 193-196, Page 5.
C5: Lines 224–226: These values correspond to the daily average TGM concentrations in each month. Clarify this.
R5:Done. Line 240, Page 6.
C6: Lines 240–242: This view is incorrect, because the differences are not statistically significant as described on line 242.
R6: This view is incorrect so we changed it. Please see Line 259, Page 7.
C7: Lines 242–244: Show the data.
R7: We supplemented the data. Lines 260-262, Page 7.
C8: Lines 249–268: Table 3 shows that there are markedly large variations in the measured Hg fluxes. I think that comparing the mean values of such data between the three examined conditions is meaningless even if the differences are statistically significant. Moreover, what are the variations in the Hg fluxes under each condition? Thus, whether significant differences in the Hg fluxes exist truly between the vegetation coverage and the vegetation removal is not clear.
R8: We think this comparison is meaningful. The mean values of such data is daily average, these averages are not constant values, but statistical values (equivalent average). You can understand it more in our methods: The mean values of two Cout and four Cin (before and after the Cout) were used to sequentially calculate the Hg fluxes between the soil and air. One Hg flux datum was obtained every 10 minutes.
C9: Figure 5: Indicate p values for these correlations. Are these correlations statistically significant?
R9: The p value indicating these correlations was statistically significant. We added Table 4. to show it. Please see Lines 331-332, Page 10.
C10: Line 313: I do not understand this heading.
R10: This title does not summarize the content of the paragraph well, so we change it to “4.1.1. Characteristics of the GEM fluxes of different terrestrial surfaces”. Line 335, Page 10.
C11: Line 315: Eliminate “however”.
R11: Done. Line 337, Page 11.
C12: Line 320: This mean value is meaningless, because the SD is markedly large.
R12: Similar to C8, we think this mean value is meaningful, the SD here is equivalent to amplitude of mercury concentration in a day.
C13: Table 4: Show the references of these data.
R13: The data references in Table 4 are mentioned in the previous article. Please see Lines 338-346, Page 11.
C14: Figure 6: Revise “ng g-1” to “ng m-3.
R14: Done.
C15: Line 377: Cite the reference.
R15: We have added the reference. Line 400, Page 12.
C16: Lines 490–491: This is incorrect, because the seasonal variations in the soil Hg concentrations are very small (see Table 2).
R16: We accepted your opinion and changed our description. Line 512, Page 14.
C17: Line 500: Eliminate “respectively”.
R17: Done. Line 522, Page 15.
Reviewer 3 Report
Dear Authors,
Thank you for this very interesting study describing the gaseous mercury exchange fluxes over vegetated and non-vegetated grassland of North-eastern China. The set of presented data is very, however, to improve changes for a more positive review, I recommend the authors to consider the following comment:
1 - Total Hg determinations should be more explored, namely tanking plant and soil sample along time for Hg analysis. This will be helpful to a better clarification on the vegetation role in Hg cycle (e.g., Hg reduction with subsequent volatilization).
2- A better explanation (graphic summary) of Hg cycle showing the effect of vegetation (tanking the bibliographic results, as well as the results of this study) would be very enriching.
You can find the attached pdf with more comments.

Author Response
Major points: (“C” means Comment, “R” means Response)
C1: 1 - Total Hg determinations should be more explored, namely tanking plant and soil sample along time for Hg analysis. This will be helpful to a better clarification on the vegetation role in Hg cycle (e.g., Hg reduction with subsequent volatilization).
R1: Thank you for your advice. In order to improve the level and integrity of the study, we will further explore the determination method of total mercury in the future. This study is a part of our study on the characteristics of grassland mercury pool. We are working to study the characteristics of mercury pool in grassland with different degradation degrees, and this study is a part of it. The role of vegetation in the mercury cycle is indeed worthy of attention. We have found some interesting phenomena in our previous studies on Leymus chinensis and Dogtail grass. For example, plants are mainly the source of mercury in the growth stage and the sink of mercury in the mature stage. This will be helpful to a better clarification on the vegetation role in Hg cycle
C2: 2- A better explanation (graphic summary) of Hg cycle showing the effect of vegetation (tanking the bibliographic results, as well as the results of this study) would be very enriching.
R2: Thank you for your advice, we added a graphical summary to better explain the mercury cycle and show the effect of vegetation. We attached the picture below.

Round 2
Reviewer 1 Report
The major point of my criticism was the experiment design and its description which is still incomplete:
- The diameter and the position of the chamber inlet?
- Sealing of the gap between the soil and the chamber?
- Are there any tests which prove that the air was not sucked through the soil?
- What is the difference between soil with "plants removed" and "bare soil"?
Without the information about these issues the paper cannot be published.
Author Response
From: Gang Zhang
School of Environment, Northeast Normal University,
Key Laboratory of Vegetation Ecology, Ministry of Education, Northeast Normal University,
Institute of Grassland Science, Northeast Normal University,
Changchun Jilin, 130117, China.
Editor in Biology
September 8 2021
Thank you for considering the revised version of our manuscript “Total Gaseous Mercury Exchange Fluxes Over Air-Soil Inter-faces in the Degraded Grasslands of Northeastern China (Ms. ID.: biology-1361200)”, for publication in Biology. We appreciate all the comments from the reviewers on the previous manuscript. We have thoughtfully considered these comments. The following
are the itemized responses to the reviewers’ comments (given in blue below). We highlighted all the
changes by using a purple font in the revised manuscript.
In addition to the reviewers’ revision suggestions, we also found some other shortcomings in the
manuscript and have corrected them in the revised manuscript. We also highlighted them by using a purple font.
Thank you very much!
Yours sincerely,
Gang Zhang behalf of the authors.
Major points: (“C” means Comment, “R” means Response)
Review #2
C1: The diameter and the position of the chamber inlet?
R1: The chamber was linked through the outlet with the mercury analyzer by Teflon™ tube (internal diameter of 0.635 cm). Thus, the diameter of the chamber inlet is slightly larger than the outer diameter of the Teflon™ tube(outer diameter of 0.8cm), so as to ensure good air tightness. Besides,the chamber inlet was positioned at approximately 5 cm above the ground on the middle of side (25cm×50cm). Please see Lines 194-200 and 202-203, Page 5.
C2: Sealing of the gap between the soil and the chamber?
R2: During measurement, put the flux chamber on the surface of the area to be measured, extract the bottom plate of the chamber, penetrate the edge of the chamber into the soil for 2cm, and cover a layer of soil around the flux chamber to improve the air tightness of the device. We added the relevant description, please see Lines 197-199, Page 5.
C3: Are there any tests which prove that the air was not sucked through the soil?
R3: Yes, of course. As mentioned in R2, embedding the flux box into the soil for two cm can effectively ensure that the outside air is not inhaled by the soil. In addition, this method is also widely used in other studies [1-4].
C4: What is the difference between soil with "plants removed" and "bare soil"?
R4: As you can see in Figure 2(b), “plants removed” just means mowing the above-ground plants and kept the under-ground plants. Thus, the difference between “the plants were mowed away” and “bare soil” is that the former has roots and the latter does not. In addition, “plants removed” is to imitate the two grassland utilization processes of grazing and mowing.
Reference
[1] Qian Yu, Yao Luo, Shuxiao Wang, Zhiqi Wang, Jiming Hao, and Lei DuanAtmos. Chem. Phys., 18, 495–509, https://doi.org/10.5194/acp-18-495-2018, 2018
[2] Ye Yuting, Sun Lumin, Zhou Liang. Chinese Journal of Ecology., 39, 3817-3828, http://doi.org/10.13292/j.1000-4890.202011.006, 2020
[3] Qian Yahui, Li Chunhui, Liang Handong. Study on surface mercury flux in Wuda Industrial Park, Inner Mongolia. Energy and environmental protection,41(10):91-98, https://doi.org/10.19389/j.cnki.1003-0506.2019.10.020, 2019
[4] Yuan, Wei; Wang, Xun; Lin, Che-Jen; Sommar, Jonas Olof; Wang, Bo; Lu, Zhiyun; Feng, Xinbin. Quantification of Atmospheric Mercury Deposition to and Legacy Re-emission from a Subtropical Forest Floor by Mercury Isotopes. Environmental science & technology., https://doi.org/10.1021/acs.est.1c02744,2021

Reviewer 2 Report
I think that the revised manuscript has been satisfactorily improved. Hence, this is acceptable for publication.
Author Response
From: Gang Zhang
School of Environment, Northeast Normal University,
Key Laboratory of Vegetation Ecology, Ministry of Education, Northeast Normal University,
Institute of Grassland Science, Northeast Normal University,
Changchun Jilin, 130117, China.
Editor in Biology
September 8 2021
Thank you for considering the revised version of our manuscript “Total Gaseous Mercury Exchange Fluxes Over Air-Soil Inter-faces in the Degraded Grasslands of Northeastern China (Ms. ID.: biology-1361200)”, for publication in Biology. We appreciate all the comments from the reviewers on the previous manuscript. We have thoughtfully considered these comments. The following
are the itemized responses to the reviewers’ comments (given in blue below). We highlighted all the
changes by using a purple font in the revised manuscript.
In addition to the reviewers’ revision suggestions, we also found some other shortcomings in the
manuscript and have corrected them in the revised manuscript. We also highlighted them by using a purple font.
Thank you very much!
Yours sincerely,
Gang Zhang behalf of the authors.
